# Identifying Personalized Metabolic Signatures in Breast Cancer

**DOI:** 10.3390/metabo11010020

**Published:** 2020-12-30

**Authors:** Priyanka Baloni, Wikum Dinalankara, John C. Earls, Theo A. Knijnenburg, Donald Geman, Luigi Marchionni, Nathan D. Price

**Affiliations:** 1Institute for Systems Biology, Seattle, WA 98109, USA; pbaloni@isbscience.org (P.B.); john.earls@isbscience.org (J.C.E.); theo.knijnenburg@isbscience.org (T.A.K.); 2Department of Oncology, Sydney Kimmel Comprehensive Cancer Center, Johns Hopkins University School of Medicine, Baltimore, MD 21205, USA; wdinala1@jhmi.edu; 3Department of Pathology and Laboratory Medicine, Weill Cornell Medicine, New York, NY 10021, USA; 4Department of Applied Mathematics and Statistics, Johns Hopkins University, Baltimore, MD 21205, USA; geman@jhu.edu

**Keywords:** breast cancer, genome-scale metabolic models, constraint-based analysis, divergence analysis, gene expression, metabolism, drug targets, personalized metabolic networks

## Abstract

Cancer cells are adept at reprogramming energy metabolism, and the precise manifestation of this metabolic reprogramming exhibits heterogeneity across individuals (and from cell to cell). In this study, we analyzed the metabolic differences between interpersonal heterogeneous cancer phenotypes. We used divergence analysis on gene expression data of 1156 breast normal and tumor samples from The Cancer Genome Atlas (TCGA) and integrated this information with a genome-scale reconstruction of human metabolism to generate personalized, context-specific metabolic networks. Using this approach, we classified the samples into four distinct groups based on their metabolic profiles. Enrichment analysis of the subsystems indicated that amino acid metabolism, fatty acid oxidation, citric acid cycle, androgen and estrogen metabolism, and reactive oxygen species (ROS) detoxification distinguished these four groups. Additionally, we developed a workflow to identify potential drugs that can selectively target genes associated with the reactions of interest. MG-132 (a proteasome inhibitor) and OSU-03012 (a celecoxib derivative) were the top-ranking drugs identified from our analysis and known to have anti-tumor activity. Our approach has the potential to provide mechanistic insights into cancer-specific metabolic dependencies, ultimately enabling the identification of potential drug targets for each patient independently, contributing to a rational personalized medicine approach.

## 1. Introduction

The physiological state of a cell is influenced by underlying metabolic processes which exhibit high degrees of heterogeneity across patients and across cells. Cancer cells reprogram their energy metabolism as is needed to meet the energy demands of proliferation and migration. The mechanisms of invasion and metastasis are complex, and mortality is mainly caused by the progression of cancer to a metastatic state [1]. Alteration of interactions between cancer cells and their microenvironment leads to diverse outcomes in the programmed behavior of the cells. Tumor cells exhibit heterogeneous metabolic profiles, with differential utilization of metabolites such as glucose, lactate, glutamine, and glycine [2]. Some of the metabolic and genetic changes that are reported in tumor cells are enhanced glycolysis, differential expression of lactate dehydrogenase A (LDH), which is linked with cancer growth and metastasis, mutations in metabolic enzymes such as isocitrate dehydrogenase 1 (IDH1), succinate dehydrogenase (SDH) and fumarate hydratase (FH) involved in initiating tumors [3]. These findings suggest that metabolism is fundamental in determining cell fate in cancer and should be explored further. Various omics measurements from diverse cancer cell lines have made it easier to study the physiological changes. Integration of these omics measurements with computational models increases the accuracy of predictions.

Transcriptome analysis provides a genome-wide snapshot of differential gene activity, providing important information about key genes that modulate metabolism at the system level. Transcriptomes are complex data types with a high degree of person-to-person heterogeneity that can obfuscate the underlying biological signal, hindering their use in practice. To partially address this issue, we have recently introduced “divergence analysis” [4], a simplified and personalized data representation that captures the departure of omics profiles from a normal reference baseline. Divergence analysis of breast cancer samples in The Cancer Genome Atlas (TCGA) [4] has been useful in measuring the degree of divergence for genes and other genomic features in cancer versus the normal baseline phenotype, as well as one cancer phenotype versus another. Divergence is a single sample property (unlike, e.g., a differentially expressed gene), and our previous work has shown that divergence encoding largely preserves biological signals and helps to remove unwanted noise from the data [4]. It is therefore helpful for data preprocessing before complex system-level analyses, including metabolic network modeling.

Combining biological data and modeling enables us to study complex interactions in a biological system. Integrating transcriptomics data onto a genome-scale metabolic network to perform network-level simulations is a useful step to regularize the data and attempt to infer metabolic states from the combined evidence of the enzymes that are expressed in the transcriptome as a whole. Many computational methods for metabolic modeling have been developed [5,6]. Genome-scale metabolic models (GSMs) provide comprehensive information about known genes, metabolites, and reactions in organisms and are useful to infer metabolic differences between conditions [7,8,9]. These models have been used to predict changing metabolic landscapes in cancers and also predict candidate drug targets and biomarkers of cancer [10,11,12,13].

The main contributions of the present work are three-fold: (1) we generate context-specific metabolic networks for 1156 cancer and normal samples by integrating their divergence profiles with a global human metabolic network reconstruction; (2) we develop a framework for identifying key metabolic and regulatory signatures and used it to classify the samples in breast cancer based on their metabolic state; (3) we perform in silico gene knockout in these 1156 context-specific metabolic networks and identify genes that can perturb the system, many of which correspond to known drug targets. Thus, our study provides a novel assessment of metabolic network analysis based on divergence encoding. Herein, we have employed this strategy for breast cancer, but our method can be extended to other cancers and metabolically perturbed diseases to identify key metabolic signatures and potential drug targets.

## 2. Results

### 2.1. Understanding Metabolic Differences in Cancer Samples Using Personalized Metabolic Networks

In this study, we used gene expression estimates, encoded as binarized divergence indicators or as transcripts per million (TPM) values, from 1156 cancer and normal samples from TCGA (https://www.cancer.gov/tcga), and integrated them with a human metabolic model (Recon 3D) [14] to obtain personalized metabolic networks for each sample. This approach allowed us to predict distinct metabolic signatures for each individual sample and classify them according to their metabolic phenotype. In this study, we have referred to personalized metabolic networks generated from divergence and transcriptome analysis as ‘divergent networks’ and ‘normalized networks’, respectively. An overview of the work done in this study is represented in Figure 1.

### 2.2. Classifying Cancer Samples Based on Their Metabolic Profile

We used genes present in human reconstruction (Recon 3D) and mapped the divergence values for solid tissue normal, primary tumor, and metastatic samples. Principal component analysis of metabolic gene expression in these samples showed two clusters, but the normal samples could not be differentiated from cancerous ones (Figure 2a). This suggested that expression profiling is not sufficient to distinguish the samples and classify them. We performed a similar analysis with TPMs and failed to identify a clear clustering of the samples (Appendix A). To obtain a better understanding of perturbations in the system, we integrated divergence and TPM values with the human metabolic model using the integrative metabolic analysis tool (iMAT) method and generated context-specific metabolic networks for 1156 primary tumor, metastatic and normal tissue samples. We observed distinct clusters for cancer (primary and metastatic) and normal samples, using fluxes measured for reactions in the context-specific networks (Figure 2b). The primary and metastatic samples were mixed in the cancer cluster. This suggested that metabolic networks were able to distinguish various phenotypes and can be used to understand mechanistic changes in the system.
(a)Class comparison: We compared the reaction fluxes for cancer and normal samples in the dataset and classified reactions in each context-specific network as active or inactive based on their flux measurement (described in the methods section). In order to identify active reactions in the context-specific networks, we used the information of reaction fluxes from all 1156 context-specific metabolic networks. If a reaction was present in the network, it was assigned a state of 1, while the remaining reactions were assigned a state of 0, indicating that they were absent in the context-specific metabolic network. Statistical analysis of active reactions in divergent networks identified 471 reactions (*p*-value < 0.05) that were significantly different in cancer versus normal. These reactions belonged to the following pathways: androgen and estrogen metabolism, bile acid synthesis, cholesterol metabolism, citric acid cycle, drug metabolism, eicosanoid metabolism, exchange reactions, fatty acid oxidation, glutathione metabolism, glycerophospholipid metabolism, glycolysis, steroid metabolism, transport, tyrosine metabolism, urea cycle, and vitamin metabolism. Appendix A represents the list of subsystems that were enriched in cancer versus normal.(b)Class discovery: We used an unsupervised machine learning method to classify the cancer samples based on their metabolic state. Using K-means clustering on the simulated reaction fluxes, we obtained four distinct clusters of cancer samples (Figure 2c). The number of clusters was determined by the elbow method; see Appendix A. The cancer clusters were then labeled from one to four, and normal tissue samples were assigned as cluster 0. We performed a detailed analysis of the four clusters to identify, if any, associations with standard clinical and pathological tumor characteristics. This analysis showed that the metabolic clusters were significantly associated with PAM50 molecular subtypes and estrogen receptor (ER) status (chi-squared *p*-value < 0.001), distinguishing the luminal A and B samples from basal-like samples, and also ER-positive and negative samples to a greater extent. Specifically, cluster two was enriched for luminal subtypes (luminal A and B) and predominantly accounted for ER-positive samples, while cluster three was enriched in basal-like and ER-negative tumors. (Figure 3 and Appendix A). The metabolic clusters of tumor and normal samples were used for identifying important reactions and subsystems in these clusters.


In addition to identifying differences between cancer and normal phenotypes, we extended our analysis to subsystems that are enriched for each identified cluster. The heatmap of enriched subsystems in cancer versus the normal samples, as shown in Figure 2d, indicated that glycine, serine, alanine, and threonine metabolism and C5-branched dibasic acid metabolism were enriched in all the clusters. Fatty acid oxidation, propanoate metabolism, citric acid cycle, and glycosphingolipid metabolism were enriched for clusters one, three, and four, whereas cluster two showed selective enrichment for peptide metabolism and exchange reactions. Androgen and estrogen metabolism, chondroitin sulfate degradation, and reactive oxygen species (ROS) detoxification were selectively enriched for cluster three samples, indicating that each cluster had a distinct metabolic profile, and we can probe their metabolic differences. We compared the reactions in each cluster with respect to those in normal samples and identified 254, 1388, 581, and 324 reactions that were significant in clusters one to four, respectively (Appendix A).

We extended our analysis to identify which types of samples were enriched in each of the clusters. We mapped information of PAM (Prediction Analysis of Microarray) 50 classifier for breast tumor intrinsic subtyping, known ER status, the triple-negative status of samples, American Joint Committee on Cancer (AJCC) stage, and vital status for samples in the cluster and obtained interesting results and performed chi-squared statistics for these clusters. We found that cluster two had a higher proportion of HER2-enriched, luminal A, and luminal B samples, whereas cluster four had a higher proportion of basal-like samples (Figure 3a). When we looked at the ER status of the samples, we observed that Cluster two had a higher proportion of ER-positive samples, and cluster four had a higher number of samples that were ER-negative (Figure 3b). Cluster two and four had a higher proportion of samples with known cases of triple-negative status (Figure 3c). For samples with known AJCC stages, we observed that cluster two had a higher proportion of samples that belonged to stage II (Figure 3d). This suggests that samples belonging to cluster two have a distinct metabolic profile and are able to distinguish tissue type and known markers of breast cancer.

To further analyze these clusters, we measured the recurrence-free survival and overall survival (deceased versus living) and observed differences between the four clusters. Based on the analysis, samples in cluster one and four had better survival than cluster two and three. So, we combined clusters one and four and clusters two and three to identify differences in survival rate. The plot of Kaplan-Meier estimates in Figure 4 shows differences between the group of clusters. This indicates that clusters with metabolic differences also had different survival and recurrence rates.

Our analyses of personalized metabolic networks showed differences in the metabolic profile of the individuals such that they could be broadly categorized into four clusters and also indicated variations at reactions level, subsystems level, and also the survival and recurrence rate. We further identified how metabolic genes contributed to these variations and probed genes that caused perturbations in the system.

### 2.3. Identifying Candidate Druggable Genes

Deletion of a set of metabolic genes from the models can either have profound effects on the system or no effect at all. In order to predict the genes that cause perturbations in the system, we carried out in silico gene deletion in our personalized metabolic networks. About 53 out of 1884 metabolic genes upon single-gene deletion had a significant effect in the system (*p* < 0.05) upon single gene deletion analysis. Table 1 represents a concise list of genes, the subsystems these genes belong to, and drug target information of these genes as reported in the Human Protein Atlas (HPA). The last column contains information on whether there are known Food and Drug Administration (FDA)-approved drugs targeting the gene.

Metabolic genes like Sterol O-Acyltransferase 1 (*SOAT1*), methylmalonyl-CoA mutase (*MUT*), and isozymes of succinate dehydrogenase (*SDHA*, *SDHB*, *SDHC*, and *SDHD*) have known FDA approved drugs that can target them. Some of the other genes identified from our analysis, like uracil phosphoribosyltransferase (*UPRT*), sphingomyelin synthase 1 (*SGMS1*), solute carrier protein (*SLC25A10*), choline/ethanolamine phosphotransferase (CEPT), phosphate cytidylyltransferase 2 (*PCYT2*), and pyruvate dehydrogenase complex component X (*PDHX*) did not have drug target information. These genes are involved in diverse metabolic processes, as indicated by the subsystems in Table 1. This analysis compared the genes in cancer versus normal samples that alter the system upon deletion and also provided information on the drugs that can target them.

Studies have shown connections between metabolism and epigenetic modifications in cancers [16,17]. For the genes identified from in silico knockout analysis (Table 1), we obtained their CpG mapping (Appendix A). Using the information of expression and methylation levels, we computed divergence for tumor samples for *SOAT1*, *SDHA*, *SDHC*, *PCYT2*, *ACOX2*, *COX4I1*, and *UQCRB*. CpGs for *SDHB* did not show divergence for tumor samples (Figure 5). 

In addition to performing systems-level analysis, we developed a method that can be used for predicting drug(s) that can be effective for each personalized metabolic network or can be used for known phenotypes in the system. In this analysis, we queried a list of genes causing an effect in the system against drug databases, and that gave us information on drugs that have a higher influence in the system. Using the drug response data from the Genomics of Drug Sensitivity in Cancer (GDSC) [18], we also identified drugs that have an influence on the cells when the genes are mutated. Table 2 lists the drugs and their targets based on the number of samples (out of 1156 total samples) that identified the genes reported from our in silico gene deletion analysis. These drugs have been tested on 1001 cancer cell lines, including 51 BRCA cell lines. The top-ranking drug, MG-132, is a proteasome inhibitor and blocks the proteolytic activity of the 26S proteasome complex. This drug has been found to be effective in inhibiting the proliferation of BRCA cells. OSU-03012 is a celecoxib and has been shown to have anti-cancer and antimicrobial activity. The drug, in combination with PDE5 inhibitors, has shown enhanced anti-tumor activity.

From our analysis, it is possible to identify drug combinations that are predicted to have more effect in cancer versus normal samples. Also, we have generated personalized drug profiles for each individual in the study, thus enabling us to predict which drug or drug combination will have a higher drug score in the individual.

## 3. Discussion

We have generated a personalized metabolic network for each sample in the study using divergence values, classified the samples into different clusters based on their metabolic profile, and identified drug/chemical moieties that can target metabolic genes identified from our analysis. We have applied these steps to breast cancer samples and identified four distinct clusters based on their metabolic profile. From the in silico gene deletion analysis, we identified metabolic genes that are altered in cancer versus normal conditions. Genes belonging to cholesterol metabolism, valine, leucine, isoleucine metabolism, as well as citric acid cycle had known FDA-approved drugs targeting them. We also carried out an *n*-of-1 analysis and identified drug responses in each sample in our study. We identified proteasome inhibitors (MG-132), COX-2 inhibitor (OSU-03012), CASP3 agonists, and an inhibitor of IGF-1R (GSK-1904529A) that targeted genes identified from gene deletion analysis of personalized metabolic networks of cancer samples. We have provided evidence that a metabolic analysis is able to provide a deeper understanding of the metabolic alterations in cancer. There are three primary findings from this study that are described below.

First, we used individual RNAseq profiles to build personalized metabolic networks to estimate candidate metabolic network states in breast cancer and control samples from TCGA (https://www.cancer.gov/tcga). We first used the divergence approach [4] to identify genes that diverged high or low based on RNAseq data normalized as transcripts per million (TPM). We then integrated this information with a genome-scale human metabolic network [14] to estimate candidate metabolic network states that would be supported by the observed high and low expression of the corresponding enzyme-encoding genes. We used the divergence method for computing values because it has the advantage of removing noise while keeping important signals in the dataset. Similar results could be obtained using continuous data, but the level of noise in the count is considerable, making it difficult to find anything useful. Whereas transcriptomic data is useful in giving us a snapshot of the extent to which genes are expressed, we need to integrate this information with computational models in order to gain mechanistic insights into the processes that are affected in the system. In this study, we leveraged our knowledge of metabolic networks and integrated divergence data to understand the metabolic landscape of breast cancer. Our workflow also allows us to carry out an *n*-of-1 analysis and generate personalized metabolic networks for each sample in the study.

Second, we identified four distinct metabolic clusters of breast cancer samples from TCGA. Cluster two had a higher proportion of samples that were HER2-enriched, luminal A and B samples, ER-positive samples, and triple-negative samples compared to clusters one, three, and four. Cluster four had a high proportion of samples that were basal-like in origin, ER-negative, and also triple-negative status. Thus, the metabolic clustering analysis gave us information on the metabolic profile of the samples that was not evident from the transcriptome data alone.

Third, from our in silico gene deletion analysis, we identified Sterol O-Acyltransferase 1 (*SOAT1*), methylmalonyl-CoA mutase (*MUT*), and isozymes of succinate dehydrogenase (*SDHA*, *SDHB*, *SDHC*, and *SDHD*) as having a significant effect (*p*-value < 0.05) in cancer as compared to normal samples. These genes had known FDA-approved drug targets that inhibited them. We also identified CpGs for metabolic genes that were most divergent in tumor samples. From our drug response analysis, we identified MG-132, a cell-permeable proteasome inhibitor, that has been known to inhibit the proliferation of BRCA cells [19,20]. This drug has been known to induce down-regulation of anti-apoptotic proteins Bcl-2 and XIAP and up-regulates the expression of pro-apoptotic protein Bax and caspase-3 in glioma cells [21]. Studies have shown that the dose of MG-132 varies based on cell type [22,23]. The drug has been shown to be effective on breast cancer cell lines and not affect the viability of normal cell lines [23]. We also identified OSU-03012 from our drug response analysis. This drug has been reported to have anti-cancer activity [24] and mediates anti-tumor effects via the inhibition of PDK1 [25]. The effect of this drug on breast cancer can be tested. PAC-1, identified from our analysis, is an activator of procaspase-3 and induces apoptosis in tumor cells [26]. Our framework provides a list of drugs that can be tested for their effectiveness in breast cancer.

Tumor cells are known to reprogram energy metabolism [27], and metabolic aberrations such as the Warburg effect are considered a hallmark of cancer [28]. Tumor cells exhibit heterogeneous metabolic profiles, with differential utilization of metabolites such as glucose, lactate, glutamine, and glycine [2]. Some of the metabolic dysregulations that have been reported in tumor cells are enhanced glycolysis, amino acid metabolism, fatty acid metabolism [3,29], which are profoundly dysregulated in cancer and have been linked with mutated genes. The bioavailability of certain metabolites, such as asparagine, has been shown to have an influence on the metastatic potential of breast cancer. These studies have shown that metabolism is altered in cancer, and it is a fundamental process that needs to be studied in-depth. Tumor cells exhibit variable metabolic profiles making it challenging to decode the heterogeneous metabolic landscape in cancer.

Our framework is generalizable and can be used for generating personalized metabolic networks that will help in categorizing the samples based on their metabolic profile and identifying drug targets that will have an effect on the system.

## 4. Materials and Methods

### 4.1. Expression Data and Divergence Analysis

We downloaded RNA-Seq data from TCGA breast cancer samples (https://www.cancer.gov/tcga), which consists of 1100 tumor (primary and metastatic) and 56 normal tissue samples. Expression counts summarized at the gene-level were retrieved from the “firehose” data portal. For metabolic model integration with gene expression and to obtain context-specific models for each sample, we used transcript per million (TPM) values that we then simplified into a ternary encoding (up, no change, down) using the divergence method [4].

Divergence analysis is a method for digitizing high-dimensional omics data into a binary or ternary representation for simplified analysis. This representation aims to remove inherent population variation in an omics sample to reveal features that are divergent from normal behavior as estimated from a baseline population. In the univariate version of divergence which was utilized here, after transforming the data to the rank space (by replacing the original RNA-Seq counts in each sample profile by their ranks within the profile) and estimating baseline regions, a gene that was differentially expressed above the baseline region was represented by 1, and one that was differentially expressed below the baseline region was represented by −1, with the remaining genes at 0. In this analysis, half of the normal breast samples were used as the reference population to estimate baseline behavior, and the divergence coding was computed for each gene for the remaining normal as well as the tumor samples. This step enabled converting the continuous gene expression value to ternary values for genes in the dataset.

### 4.2. Integration of Expression Data to Generate Personalized Metabolic Networks

For our analysis, we used the latest genome-scale reconstruction of known human metabolism, Recon3D, which is a multi-compartment model consisting of 10,600 reactions, 5835 metabolites, 2248 metabolic genes as well as 102 subsystems [14]. Gene-protein-reaction (GPR) associations in the genome-scale metabolic models (GEMs) were used for integrating omics information with the models. TPM and divergence values were calculated for RNA-Seq data from TCGA. These values were integrated with the Recon3D model [14] using iMAT [30] (Appendix A). In this way, we generated 1156 context-specific metabolic networks and predicted a reaction rate (‘flux’) for each reaction in the network. Reactions related to biomass synthesis and ATP synthase were considered as core reactions and retained for the generation of context-specific metabolic networks. We performed flux balance analysis (FBA) using COBRA toolbox v 3.0 [31] and evaluated flux distribution using linear programming (LP) solvers [32], using an objective function that was previously reported for cancer cells [33]. We used Ham’s media composition [14] for constraining exchange reactions in the context-specific networks. Using fastFVA [34], the flux values for reactions supporting 90% of biomass production were calculated and used to classify reactions as active or inactive in the context-specific networks. The workflow represented in Figure 1 provides an overview of analyses performed. COBRA toolbox v3.0 was implemented in MATLAB R2018a, and academic licenses of Gurobi optimizer v7.5 and IBM CPLEX v12.7.1 were used to solve LP and MILP problems in this study.

### 4.3. Classification of Context-Specific Metabolic Networks into Metabolic Subgroups

We carried out flux variability analysis for all context-specific metabolic networks using fastFVA [34]. Maximum flux values for reactions were used for unsupervised machine learning methods to identify metabolic clusters of cancer samples (Appendix A). K-means clustering was performed in R using the package cluster and factoextra for cluster and visualization. We computed the distance matrix using Pearson correlation. In order to ascertain the optimal number of clusters, we used the “elbow method” that takes into account the total within-cluster sum of squares (wss). Appendix A represents the curve obtained for wss according to the number of clusters k. We distinctly observed four clusters for cancer metabolic networks using K-means clustering [35]. The cancer clusters were then labeled from one to four, and normal tissue samples were assigned as cluster zero for our analysis.

Using Fisher’s exact test, we identified reactions that were statistically significant in cancer versus normal and also in clusters one to four (cancer clusters). The list of active reactions in each cancer cluster was compared with normal tissue samples to determine subsystems that were enriched in each cluster. We also examined the clusters with important phenotypes such as PAM50, ER status of the individual, triple-negative status, tissue source site, year of initial pathological diagnosis, and pathological state (Appendix A).

### 4.4. Identifying Target Genes in the Context-Specific Networks

We performed in silico gene deletion analysis using the singleGeneDeletion function in COBRA toolbox [31]. The total number of genes in the Recon3D model was 2248, of which 1883 were unique genes. We deleted genes in the context-specific metabolic networks one at a time and measured the ratio of the growth rate of the knockout model versus the wild type model. Genes with a growth rate ratio (grRatio) < 0.9 were considered to have an impact on the system and were used as input for drug target prediction. A grRatio of 0.9 suggested that the knockout model was able to attain 90% of its growth compared to the original model. A Wilcoxon rank-sum test was carried out to identify genes that had a significant effect on the system upon knockout in cancer versus normal context-specific networks. Information from the Human Protein Atlas [15] and the Pathology Atlas [36] was used for biological annotation of these genes and identification of these genes as FDA approved drug targets or potential drug targets based on HPA.

### 4.5. Metabolic Genes Divergent in the Expression and Methylation Space

The data was retrieved from TCGA data portal (https://www.cancer.gov/tcga). The gene to CpG mapping was obtained from the Illumina 450k annotation database. The mapping indicates some CpGs as being mapped in proximity to the promoter region of the corresponding gene. The methylation levels are the beta values in the (0,1) range—i.e., the proportion of methylation at each CpG. Using the gene expression and CpG methylation data, we computed Spearman rank correlations for breast TCGA normal, and tumor data for CpGs mapped to each gene. The data for correlation analysis is shown in Appendix A. Tumor samples in common between the expression data and the methylation data were selected. Divergence for tumor samples was calculated separately for both RNA-seq gene expression data (using normal gene expression as a baseline) and for methylation (using normal methylation as a baseline).

### 4.6. Drug Target Identification for Genes Shortlisted from Metabolic Networks

We calculated statistical associations between in vitro drug sensitivity data and the personalized target gene sets, as shown in Appendix A. Specifically, we used the drug response data from Genomics of Drug Sensitivity in Cancer (GDSC) [18], which contains IC50s for 265 anti-cancer drugs across 1001 cancer cell lines, including 51 BRCA cell lines. GDSC also included a genomic and molecular characterization of these 1001 cell lines. We used the binarized mutation data of more than 19,000 genes, including only protein-changing mutations [18,37]. For each of the 1156 samples, we created a binary vector across the 1001 GDSC cell lines indicating whether a cell line has at least one mutated gene in the essential gene set of the sample under investigation. A Spearman rank correlation coefficient was computed between the binary vector and the continuous IC50 drug response values for each of the drugs (*n* = 265). We selected drugs for which at least one of the samples the *p*-value is smaller than 1 × 10^3^ (uncorrected). Negative correlation coefficients indicate that mutated cell lines (i.e., those that have mutations in metabolic genes) are more sensitive (low IC50) to a drug.

### 4.7. Statistical Analysis

Fisher’s exact test was the statistical method for identifying significant active reactions from the models. For identifying differentially expressed genes in cancer versus normal, we used a Wilcoxon rank-sum test. To account for the multiple testing in these analyses, we calculated the Benjamini–Hochberg (BH) False Discovery Rate correction and a BH-FDR < 0.05 was considered as significant.

### 4.8. Software

The R/Bioconductor package ‘divergence’ was used for the divergence computation. We used the COBRA toolbox v3.0 [31] in MATLAB 2018a for analyzing the metabolic networks. Academic licenses of the Gurobi optimizer v7.5 and IBM CPLEX v12.7 were used to solve LP and MILP problems. PCA and K-means clustering were done using R 3.5.0 (codename “Joy in Playing”). For K-means clustering, we used the package ‘cluster’ and ‘factoextra’ for clustering and visualization.

## Figures and Tables

**Figure 1 metabolites-11-00020-f001:**
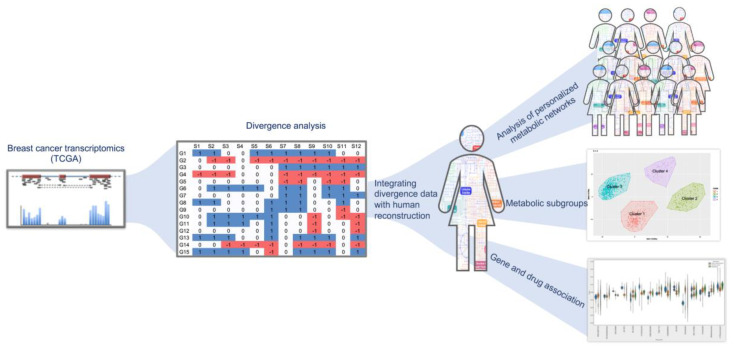
Overview of the study design. The breast cancer expression dataset from The Cancer Genome Atlas (TCGA) was converted to ternary format using divergence analysis (shown in the middle panel) [4]. The divergence values were integrated with human reconstruction and pruned to obtain personalized metabolic networks. The right side of the figure panel shows the identification of metabolic subgroups in the samples using unsupervised clustering. From our analysis, we identified important reactions and genes in cancer versus normal and used this information to associate drugs that can target them (bottom panel on the right).

**Figure 2 metabolites-11-00020-f002:**
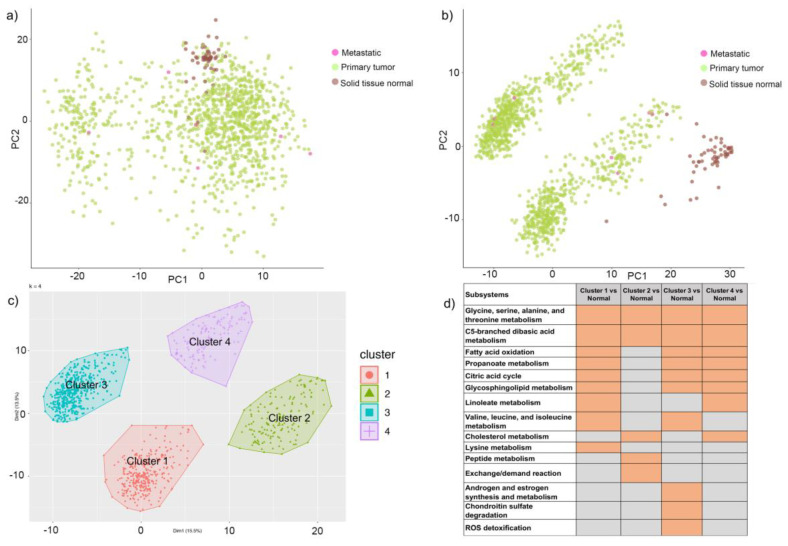
Cluster analysis of genes and reactions. (**a**) Principal component analysis (PCA) of metabolic genes (divergent values) of 1156 breast cancer samples from TCGA. Samples are colored as brown, green, and pink based on normal, primary or metastasis phenotype, respectively. (**b**) PCA of 1156 samples clustered based on metabolic reaction fluxes and colored with respect to sample type. Samples are colored as brown, green, and pink based on normal, primary, or metastasis phenotype, respectively. (**c**)Four clusters of cancer samples indicating distinct metabolic profiles. The clusters have been labeled as 1, 2, 3, and 4. (**d**)Heatmap representing enriched subsystems for each cluster when compared to normal samples. Orange fields indicate significant subsystems with a *p*-value < 0.05, and the gray fields indicate non-significant subsystems with a *p*-value > 0.05.

**Figure 3 metabolites-11-00020-f003:**
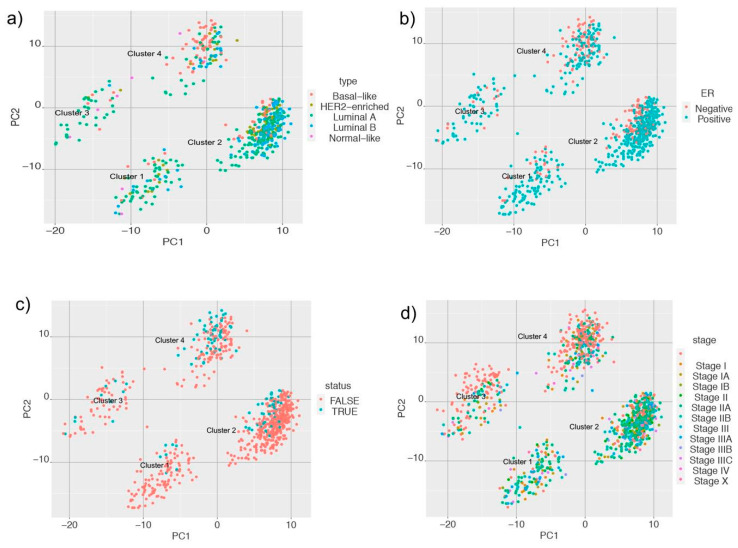
Principal component analysis (PCA) plots of metabolic clusters considering (**a**) Prediction Analysis of Microarray 50 (PAM50); (**b**) estrogen receptor (ER) status; (**c**) triple-negative status; and (**d**) American Joint Committee on Cancer (AJCC) stage of the samples.

**Figure 4 metabolites-11-00020-f004:**
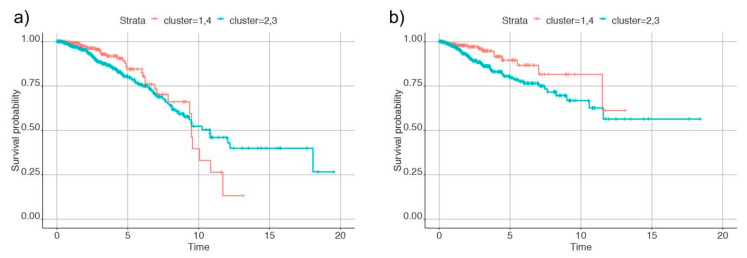
Plots of Kaplan-Meier estimates for (**a**) overall survival and (**b**) recurrence of cancer in the individuals. Clusters 1 and 4 are denoted by the red line and clusters 2 and 3 by the blue line.

**Figure 5 metabolites-11-00020-f005:**
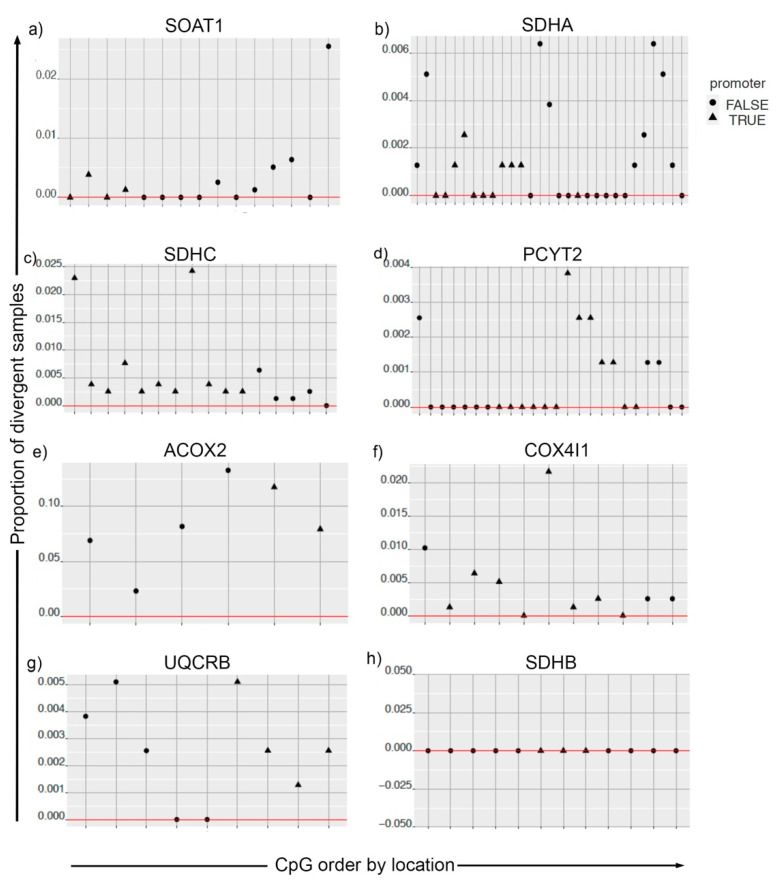
Divergence analysis for methylation pattern of metabolic genes in cancer and solid tissue normal samples. For plots (**a**–**h**), the CpGs on the x-axis are ordered by their location. The y-axis is the proportion of tumor samples that are divergent both in the expression space and methylation space simultaneously. CpGs mapped to promoter regions are indicated by the shape of the points.

**Table 1 metabolites-11-00020-t001:** List of genes identified as important from in silico gene knockout analysis and mapped to their subsystems and known drug target information from Human Protein Atlas (HPA). Drug target information for the genes is provided in the last column (FDA-approved drugs or potential drug target) using the information from HPA [15].

Subsystem	Gene	Drug Target
Cholesterol metabolism	SOAT1	FDA approved
Valine, leucine, and isoleucine metabolism	MUT	FDA approved
Citric acid cycle	SDHA, SDHB, SDHC, SDHD	FDA approved (SDHD), Potential drug target
C5-branched dibasic acid metabolism	SUCLA2, SUCLG1, SUCLG2	Potential drug target
Lysine metabolism	DLD, DLST	Potential drug target
Oxidative phosphorylation	ATP5 family, COX family, UQCR family, CYC1, CYTB	Potential drug target
Pyrimidine synthesis	UPRT	
Sphingolipid metabolism	SGMS1	
Transport, mitochondrial	SLC25A10	
Glycerophospholipid metabolism	CEPT1, PCYT2, PDHX	

**Table 2 metabolites-11-00020-t002:** Information about drugs ranked based on their influence on genes identified from our in silico analysis. The target information and the number of samples in which these genes are observed are also indicated in the table.

Drug	Brand Name	Target	#Significant Samples (out of 1156)	Cohort
MG-132		Proteasome	599	BRCA
OSU-03012		PDPK1 (PDK1)	474	All cell lines
PAC-1		CASP3 agonist	94	All cell lines
GSK-1904529A		IGF1R	89	All cell lines
PF-562271		FAK	31	All cell lines
QS11		ARFGAP	28	All cell lines
Trametinib	Mekinist	MAP2K1 (MEK1), MAP2K2 (MEK2)	28	All cell lines
XMD11-85h		BRSK2, FLT4, MARK4, PRKCD, RET, SPRK1	23	All cell lines
(5Z)-7-Oxozeaenol		MAP3K7 (TAK1)	14	All cell lines
GSK-650394		SGK3	12	All cell lines
Tipifarnib	Zarnestra, IND58359, R115777	Farnesyl-transferase (FNTA)	12	All cell lines
Vinorelbine	Navelbine	Microtubules	8	All cell lines
5-Fluorouracil		DNA antimetabolite	5	All cell lines

## Data Availability

The data presented in this study are available in article and Appendix A.

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
