# Peer review of "Identifying Personalized Metabolic Signatures in Breast Cancer"

_metabolites, 2020, doi:10.3390/metabo11010020_

Round 1

Reviewer 1 Report

In this study, Baloni et al used divergence analysis on gene expression data of 1156 breast normal and tumor samples from The Cancer Genome Atlas (TCGA) and integrated this information with a genome-scale reconstruction of human metabolism to generate personalized, context-specific metabolic networks. This is an interesting study and the manuscript is clearly written. It provides useful information for the understanding of the metabolic differences between interpersonal heterogeneous cancer phenotypes.  As the connection between metabolism and epigenetic modifications plays an important role in the dysregulation of gene expression in cancers, I would suggest to show the correlation of metabolic and epigenetic signatures in normal and tumor samples.

Reviewer 2 Report

This is a well-written manuscript and I thoroughly enjoyed reading it. It would be great if the authors elaborate on how these findings could be used in the real-world/clinic. Below are some additional comments that need to be addressed.

1) Line 3: correct spelling of cancer

2) Please update Figure 2A, 2B, and the figure legend.

3) Include references for Table 1

4) Table 1 please mention what does the drug treat.

5) Line 222, the authors mention they can predict drug effects, but it is not clear how and using what? please elaborate

6) Line 283, please check the reference 16. This might be a wrong reference

7) MG-132 is one of the potential anti-cancer drugs identified in this study. This drug is highly toxic to the normal cells if incubated for a long time at a particular concentration. I encourage the authors to discuss how they are going to filter a drug for its anti-cancer application based on what is currently known about that particular drug.
